# RANGER: REPOSITORY-LEVEL AGENT FOR GRAPH-ENHANCED RETRIEVAL

## ABSTRACT

General-purpose automated software engineering (ASE) includes tasks such as code completion, retrieval, repair, QA, and summarization. These tasks require a code retrieval system that can handle specific queries about code entities, or *code entity queries* (for example, locating a specific class or retrieving the dependencies of a function), as well as general queries without explicit code entities, or *natural language queries* (for example, describing a task and retrieving the corresponding code). We present **RANGER**, a repository-level code retrieval agent designed to address both query types, filling a gap in recent works that have focused primarily on code-entity queries. We first present a tool that constructs a comprehensive knowledge graph of the entire repository, capturing hierarchical and cross-file dependencies down to the variable level, and augments graph nodes with textual descriptions and embeddings to bridge the gap between code and natural language. RANGER then operates on this graph through a dual-stage retrieval pipeline. Entity-based queries are answered through fast Cypher lookups, while natural language queries are handled by MCTS-guided graph exploration. We evaluate RANGER across four diverse benchmarks that represent core ASE tasks including code search, question answering, cross-file dependency retrieval, and repository-level code completion. On CodeSearchNet and RepoQA it outperforms retrieval baselines that use embeddings from strong models such as Qwen3-8B. On RepoBench, it achieves superior cross-file dependency retrieval over baselines, and on CrossCodeEval, pairing RANGER with BM25 delivers the highest exact match rate in code completion compared to other RAG methods.

## 1 INTRODUCTION

Retrieving relevant code snippets, functions, and classes from large repositories is central to modern software engineering, as the quality of retrieved context underpins downstream tasks for AI agents and large language models, including code generation, patch generation, automated program repair, and intelligent code completion. While retrieval over natural language has seen rapid progress (Karpukhin et al., 2020; Izacard et al., 2022), code retrieval remains substantially more challenging. Unlike natural language, code often contains long-range and multi-hop dependencies (Allamanis et al., 2018a), where the semantics of a program may depend on variables, function calls, or imports that appear far apart in the source. These properties render simple flat indexing insufficient for code retrieval, motivating the use of graph databases (Liu et al., 2024d) and multi-hop reasoning to capture cross-file relationships, call graphs, and dependency chains (Guo et al., 2022; Ye et al., 2022).

An additional challenge in code retrieval arises from query diversity. *Code-entity queries* ask questions about specific code-entities (e.g., "What are the dependencies of `Calculator` class?"). In contrast, *natural language queries*, describe behaviors or constraints without naming symbols (e.g., "Where do we implement addition?"). Natural language queries (Mastropaolo et al., 2021; Zhang et al., 2022) are particularly difficult due to the semantic gap between natural and symbolic languages (Husain et al., 2019; Gu et al., 2021b; Liu et al., 2024f; Li et al., 2025), as well as embedding anisotropy and hubness in code representations (Li et al., 2022; Gong et al., 2023).

Graph retrieval offers a promising direction by enabling multi-hop traversal while preserving hierarchical relationships, in contrast to flat index RAG (Zhong et al., 2024; Wang et al., 2023a).

By modeling the repository as a graph, where nodes correspond to code entities and edges encode hierarchical or dependency links, GraphRAG can resolve queries that require following transitive dependencies, such as tracing a function call across multiple intermediate layers or modules. However, current graph-based code retrieval methods tend to perform well on code-entity or structure-aware queries, but lack dedicated support for open-ended natural language queries (Cao et al., 2024; Ouyang et al., 2024; Liu et al., 2024e).

To address these challenges, we develop an efficient knowledge graph construction procedure together with a Monte Carlo Tree Search (MCTS)-based graph traversal algorithm. Using an agentic architecture, we integrate the knowledge graph with MCTS to enable a dual-stage retrieval system capable of handling both symbolic code-entity queries and natural language queries. Our key contributions are as follows:

- **Efficient Knowledge Graph Construction for Code Retrieval:** A tool to transform Python repositories into an information-rich knowledge graph that captures hierarchical and cross-file dependencies by parsing abstract syntax trees (AST). To mitigate the semantic gap between natural and symbolic coding languages, we augment graph nodes with textual descriptions of code entities and their corresponding embeddings.

- **Monte Carlo Tree Search-Based Graph Traversal Algorithm:** A graph traversal algorithm inspired by Monte Carlo Tree Search that balances exploration and exploitation. Starting from a source node, it quickly expands to promising candidates using a bi-encoder. During the simulation phase, a cross-encoder computes reward scores for visited nodes. Over time, rollouts uncover the most relevant node for retrieval.

- **Router Retrieval Agent:** A dual-stage retrieval pipeline that routes queries by type. Code-entity queries are resolved through fast Cypher lookups on the graph database, while natural language queries fall back to the MCTS-based graph traversal algorithm.

## 2 RELATED WORK

**Code LLMs and Retrieval-Augmented Generation**    Early neural models for source code established that structure-aware encoders using Abstract Syntax Tree (AST) paths (e.g., code2vec (Alon et al., 2019b), code2seq (Alon et al., 2019a)) or graph neural networks (Mou et al., 2016) (Allamanis et al., 2018b) could outperform lexical approaches. Subsequently, Transformer-based pretraining became the dominant paradigm, with models like Codex (Chen et al., 2021), CodeGen (Nijkamp et al., 2022), CodeLlama (Roziere et al., 2023), StarCoder2 (Lozhkov et al., 2024), and DeepSeek-Coder (Guo et al., 2024) demonstrating strong performance on function- and file-level tasks. However, these models condition on local context and struggle to incorporate the cross-file dependencies essential for reasoning in large repositories.

Early retrieval-augmented generation (RAG) systems such as RECODE (Wang et al., 2023b), REDCODER (Parvez et al., 2021), and TreeGen (Sun et al., 2020) injected external code snippets into prompts. These methods treated code as flat text, relying on lexical or vector similarity, which hindered their ability to reason across multiple files. While later work improved recall, it remained snippet-centric and failed to model the typed, multi-hop relationships that connect definitions and uses across a codebase.

**Natural Language Code Search**    Natural language–based code search has been extensively studied, beginning with large-scale benchmarks such as CodeSearchNet (Husain et al., 2019), which enabled systematic evaluation of neural retrieval models. Subsequent work enriched code embeddings with structural signals, including program dependency graphs Wang et al., 2020, (Chen et al., 2024) and variable flow graphs (deGraphCS, Zhang et al., 2021), while efficiency-focused methods like ExCS (Zhang et al., 2024a) improved scalability through offline code expansion. More recently, repository-level approaches employ multi-stage pipelines that integrate commit metadata with BERT re-rankers (Sun et al., 2025) or translate natural language queries into domain-specific query languages (Liu et al., 2025). In parallel, query reformulation (Ye & Bunescu, 2018) and LLM-driven paraphrasing(Wang et al., 2023c) highlight the central challenge of aligning vague natural descriptions with precise code identifiers, especially in large and evolving repositories.

**Graph-Based Retrieval and Agentic Frameworks** Graph-centric methods address structural limitations by explicitly encoding relationships like definitions, references, and calls, but they differ significantly in scope, persistence, and query support. Some approaches build local graphs, for instance, GraphCoder (Liu et al., 2024e) creates Code Context Graphs for snippets but omits cross-file links. CatCoder (Pan et al., 2024) constructs on-the-fly type-dependency subgraphs for statically-typed languages, sacrificing the persistent, long-range relationships needed at repository scale.

Repository-scale graphs improve coverage but introduce trade-offs. RepoGraph (Ouyang et al., 2024) separates definitions and references into distinct nodes with basic invoke/contain edges, which creates redundancy and lacks semantic embeddings for text-code alignment. CoCoMIC (Ding et al., 2024) models cross-file relations at the file level through imports rather than direct function-to-function edges, constraining multi-hop precision. RepoFuse (Cao et al., 2024) uses Jedi-based analysis to build an in-memory graph of imports, inheritance, and calls but focuses on rule-based neighbor capture for completion. Similarly, DraCo (Zhang et al., 2024b) constructs a fine-grained, variable-level dataflow graph with typed edges (`Assigns`, `Refers`, `Typeof`) but remains specialized for code completion tasks. CodeGraphModel (Tao et al., 2025) integrates a repository graph into an LLM via a graph-adapter but relies on lightweight analysis and a simple retrieval method based on entity extraction and string matching, limiting its support for non-entity and multi-hop queries.

A growing line of work couples LLMs with code graphs in agentic frameworks. LocAgent (Chen et al., 2025) converts entire codebases into directed graphs and exposes tools like `SearchEntity` and `TraverseGraph`, but its comprehensive traversals can be computationally expensive without a persistent graph database. OrcaLoca (Yu et al., 2025) uses priority-based scheduling and in-memory NetworkX graphs derived from ASTs but acknowledges that its incomplete reference analysis can miss semantic dependencies. CodexGraph (Liu et al., 2024d) bridges LLM agents with graph databases for structure-aware retrieval, but its workflows often rely on explicit identifiers, making purely natural language queries challenging. MCTS-based agents like LingmaAgent (Ma et al., 2024) explore code graphs with LLM-based reward estimation, while related variants such as RTSoG (Long et al., 2025) and REKG-MCTS (Zhang et al., 2025) apply similar strategies to document and text knowledge graphs, but the repeated high-fidelity LLM scoring incurs significant inference cost and can introduce nondeterminism. These trends highlight a need for agents that combine persistent, semantically augmented graphs with cost-aware planning to balance accuracy and efficiency.

This work presents **RANGER**, a repository-level retrieval agent that integrates persistent graph construction with query-type–aware retrieval. A repository-wide knowledge graph is built through AST parsing and enriched with semantic descriptions and embeddings. At query time, RANGER first converts the input into a Cypher query over this graph. For *code-entity queries*, these Cypher lookups typically suffice for direct resolution. For *natural language queries*, which often fail to return direct matches, RANGER invokes an MCTS-based graph exploration that combines bi-encoder expansion with selective cross-encoder scoring. This dual-path design enables efficient handling of both symbolic and natural language queries, overcoming the limitations of flat embedding indices and gaps of prior graph-based retrieval methods.

## 3 METHODOLOGY

### 3.1 OVERALL ARCHITECTURE

We propose a retrieval agent capable of processing both *natural language* and *code-entity* queries for code retrieval. As mentioned earlier, natural language queries are challenging due to the semantic gap between textual descriptions and code embeddings (Gu et al., 2021a; Husain et al., 2019).

As illustrated in Figure 1, the system uses a two-stage pipeline with an *offline indexing* stage for repository preprocessing and graph construction and an *online query* stage for retrieval and reasoning with RANGER. In the offline stage, a code repository is parsed into an entity graph stored in a graph database (e.g., Neo4j). This includes AST parsing to build the knowledge graph, LLM-assisted description generation for components and modules, and embedding computation for those descriptions.

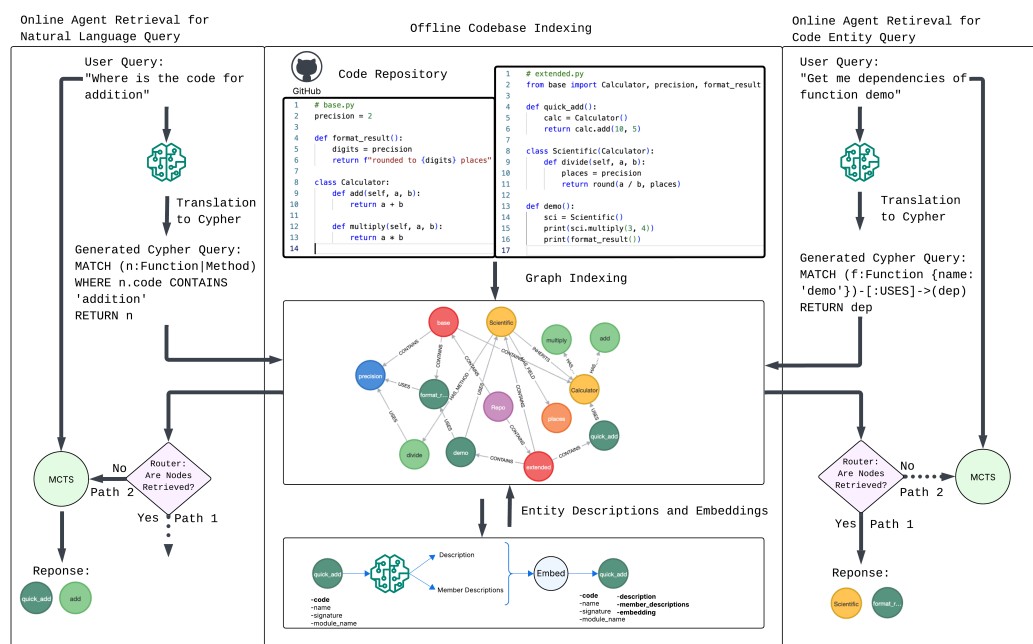

Figure 1: RANGER system architecture illustrated through a simple two-file repository example containing `base.py` and `extended.py`. The offline stage constructs a comprehensive knowledge graph from code repositories through AST parsing, LLM-assisted semantic description generation, and embedding computation. In the online stage RANGER first translates user queries into Cypher queries using an LLM. For *code-entity queries*, this Cypher query is sufficient and provides fast retrieval (Path 1). If retrieval results from the graph database return `None`, often in case of *natural language queries*, the system invokes MCTS-based graph exploration (Path 2) to generate the final response.

In the online stage, RANGER first converts the user query into a Cypher statement via zero-shot LLM prompting (prompt in the Appendix). The Cypher query retrieves relevant code entities from the graph database. For *code-entity queries*, these results typically suffice for direct response generation (Path 1). In contrast, *natural language queries* often do not match directly and return `None`. In such cases, the agent follows Path 2, invoking a Monte Carlo Tree Search (MCTS) based graph exploration to iteratively localize the most relevant code snippets. This dual-path design allows RANGER to handle both query types robustly. The following subsections detail the components of this architecture.

## 3.2 CODE PARSING AND KNOWLEDGE GRAPH CREATION

The repository-level knowledge graph is constructed through a two-stage process that first builds isolated file-level graphs and then stitches them into a unified repository-level graph. This design ensures that intra-file structures are captured accurately before resolving complex inter-file dependencies. An illustrative example of this process, using the two-file repository from Figure 1, is provided in Section A.2.

**Stage 1: File-level parsing.** Each file is processed using the `tree-sitter` library (Brunsfeld et al., 2013), which produces a detailed Abstract Syntax Tree (AST). This contrasts with existing systems (Cao et al., 2024; Liu et al., 2024d) that rely on Python-specific tools like Jedi or Parso. We traverse the AST to extract key code entities and relationships, which are organized into an intermediate JSON object serving as a decoupled transfer representation. A database-specific ingestion component then converts these objects into nodes and edges in the graph database. This separation allows new programming languages to be supported by modifying only the AST parser, and new graph backends by updating only the ingestion module. The node types include `Module`, `Class`,

`Function`, `Method`, `Field`, and `GlobalVariable`, offering finer granularity than related approaches (Ma et al., 2024; Chen et al., 2025). Within each file, structural edges are immediately established, including `CONTAINS` edges from a `Module` to its classes and functions, `HAS_METHOD` edges from a `Class` to its methods, and `INHERITS` edges to represent class inheritance. To handle unresolved dependencies, temporary `Import` nodes are created, pointing to entities outside the current file. Unlike existing approaches such as the Code Graph Model (Tao et al., 2025), which applies lightweight semantic analysis, or OrcaLoca (Yu et al., 2025), which omits static analysis, this step explicitly preserves placeholders for cross-file references.

**Stage 2: Repository-level consolidation.** After all files are parsed, the system resolves the temporary `Import` nodes. Each `Import` node is matched to its corresponding entity (`Class`, `Function`, `Module`, etc.) elsewhere in the repository, and all incoming edges are redirected to the resolved node. This "stitching" step ensures that cross-file dependencies are explicitly represented, yielding broader coverage than prior approaches such as the lightweight cross-file analyses in the Code Graph Model (Tao et al., 2025) or the limited function-call tracking in Lingma Agent (Ma et al., 2024). Once redirected, redundant `Import` nodes are deleted. The result is a repository-level knowledge graph that completely represents both intra-file structure and inter-file dependencies.

### 3.3 LLM-Assisted Semantic Description and Embedding

After constructing the knowledge graph, we add semantic attributes by generating natural language descriptions for each code entity with an LLM using a hierarchical bottom up procedure. Following Code2JSON (Singhal et al.), each entity receives two descriptions, a high level purpose summary and a granular member level summary of important variables and behaviors. For small entities such as functions and methods, whose source code fits within the context limit of the LLM, we generate both descriptions directly from code, while for larger entities such as modules and large classes we compose them from precomputed member summaries. We then concatenate the two descriptions, encode them into a vector embedding, and store the text and embedding as node attributes. Prompts are in Appendix A.9.

### 3.4 MCTS-Based Graph Traversal Algorithm

To efficiently search the code knowledge graph, we use Monte Carlo Tree Search (MCTS) to balance retrieval efficiency and accuracy. A bi-encoder guides exploration and a cross-encoder scores only the most promising candidates, which focuses computation where expected relevance is highest (Wu et al., 2019). The process, formalized in Algorithm 1, consists of Selection, Expansion, Simulation, Backpropagation, and a final Extraction stage.

**Selection.** The selection phase balances exploration (searching new parts of the graph) with exploitation (focusing on paths that have previously yielded high rewards). Starting from the root of the search tree, we recursively select the child node with the highest Upper Confidence bound for Trees (UCT) score, defined as: $\text{UCT}(v) = \dfrac{R_v}{\max(1, N_v)} + c\sqrt{\dfrac{2\ln\max(1, N_{\text{parent}(v)})}{\max(1, N_v)}}$ where $R_v$ is the total reward of a node $v$, $N_v$ is its visit count, and $c$ is an exploration parameter. We continue until a leaf is reached. If that leaf is fully expanded, we backtrack to the nearest ancestor with unexpanded neighbors.

**Expansion.** Once a leaf node is selected, the search tree is expanded by adding its neighbors from the code graph as child nodes. To guide this expansion, the bi encoder ranks all neighbors based on the cosine similarity of their embeddings with the query embedding. The top-$k$ most similar and previously unvisited neighbors are then added to the search tree. This bi-encoder driven expansion serves as a fast and effective heuristic for candidate generation.

**Simulation.** This stage evaluates the relevance of newly expanded nodes. Unlike MCTS in adversarial games (Silver et al., 2017), where rollouts simulate sequences of actions to a terminal state, our retrieval task lacks a discrete win/loss outcome. A random traversal from a node is

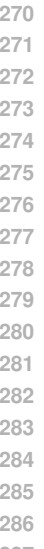

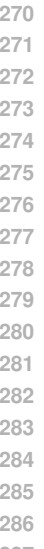

Figure 2: **The Monte Carlo Tree Search–based graph traversal algorithm.** The diagram depicts iterations 2 and 3 of a Monte Carlo Tree Search–based graph traversal on the simple two-file code repository knowledge graph from Figure 1 in response to the query "Where is the code for addition?" Iteration 2 expands the `base` module (adding `precision` and `Calculator`), simulates their rewards, and back-propagates values to update selection scores. Iteration 3 then adds the method node `add` and function node `quick_add`, both of which yield high rewards and are ultimately selected during the extraction and ranking phase as the answer to the user's query.

ill-suited for determining its relevance to a query. Therefore, we redefine the simulation step as a direct relevance evaluation using a cross-encoder. The query and the node's semantic content are used as input to the cross-encoder, which produces a precise relevance score. This score serves directly as the reward for the node. To maximize throughput, evaluations are processed in batches.

**Backpropagation.** After evaluation we propagate the reward up the tree. For every node on the path to the root we increment its visit count ($N_v$) and add the reward to its total ($R_v$). This update guides subsequent selection toward promising regions of the code graph.

**Extraction** After a predefined number of iterations the search terminates and we extract a ranked list of relevant code nodes. The final score for each visited node is $s(v) = \alpha \cdot \dfrac{R_v}{\max(1, N_v)} + (1 - \alpha) \cdot \text{sim}(E_q, E_v)$ which balances the learned MCTS reward with the initial bi encoder similarity to yield a robust final ranking.

## 4 EXPERIMENTS

We evaluate RANGER on four diverse datasets spanning both *code-entity* and *natural-language* query types and three practical scenarios covering repository-level code retrieval, code completion, and question answering.

### 4.1 NATURAL LANGUAGE QUERY BASED RETRIEVAL

#### 4.1.1 DATASETS & SETUP

**CodeSearchNet** Challenge (Python split) consists of 99 natural language queries with expert relevance annotations over a large corpus of Python functions (Husain et al., 2019). We select 70

| Metric | RANGER | Code Embedding | | Text Embedding | |
|---|---|---|---|---|---|
| | (MCTS iter) | CodeT5-110M | Qwen-3-8B | Qwen-3-8B | mxbai[1] (335M) |
| **CodeSearchNet Dataset** | | | | | |
| NDCG@10 | **0.786** (200) | 0.419 | 0.725 | 0.701 | 0.664 |
| Recall@10 | **0.911** (200) | 0.643 | 0.891 | 0.856 | 0.847 |
| **RepoQA Dataset** | | | | | |
| NDCG@10 | **0.741** (500) | 0.718 | 0.722 | 0.709 | 0.706 |
| Recall@10 | **0.890** (500) | 0.810 | 0.850 | 0.810 | 0.810 |

Table 1: **Performance comparison on CodeSearchNet and RepoQA.** RANGER consistently outperforms baseline embedding models across datasets. Iteration counts are shown in parentheses. Best baseline results are bolded.

repositories with the highest query counts, build knowledge graphs from corresponding commits, and prune nodes not present in the official corpus to align with ground truth annotations.

**RepoQA** originally evaluates long context code understanding via the Searching Needle Function task where multiple functions are provided to an LLM as context along with a function description and the LLM must return the corresponding function. To facilitate our evaluation we modify the task so that all functions become our corpus and the function description becomes our natural language query (Liu et al., 2024b).The function description includes Purpose, Input, Output, and Procedure fields, but to better reflect realistic queries, we use only the Purpose field as the natural language query. We use the Python split with ten repositories and ten descriptions per repository.

For both datasets we generate text descriptions and embeddings as detailed in Section 3 and run the MCTS stage for retrieval.

### 4.1.2 BASELINES AND RESULTS

We compare to two vector search baselines. The first uses raw code embeddings indexed directly from corpus chunks. The second uses embeddings of LLM generated semantic descriptions. This isolates MCTS gains beyond gains from descriptive text.

Table 1 reports NDCG@10 and Recall@10 on CodeSearchNet and RepoQA. RANGER improves both metrics over the baselines and also exceeds retrieval with Qwen-3-8B (Wang et al., 2025) embeddings which are currently top ranked on the MTEB leaderboard (Muennighoff et al., 2022). The improvements stem from the use of cross-encoder scoring, which provides higher accuracy than bi-encoder similarity but is too expensive to apply exhaustively. RANGER addresses this with an MCTS-guided traversal, where the bi-encoder expands promising graph paths and the cross-encoder is applied only to high-value candidates. This selective application preserves the accuracy benefits of cross-encoders while keeping retrieval computationally tractable.

Figure 3 shows that NDCG@10 and Recall@10 improve steadily with additional MCTS iterations before the rate of improvement slows. The curves exhibit clear knees that indicate the optimal iteration range for practical deployment, balancing retrieval quality with computational cost.

### 4.2 CODE-ENTITY QUERY BASED RETRIEVAL

### 4.2.1 DATASET & SETUP

**RepoBench** (Liu et al., 2024c) evaluates repository-level retrieval via RepoBench-R, where the task is selecting the most relevant cross-file snippet to support next-line prediction. We use the Python v1.1 split and restrict to repositories with at least five data points (430 repositories). The prompt provides an incomplete in-file chunk with code entities, which RANGER converts into Cypher queries to retrieve cross-file dependencies before ranking (example in Appendix). Because commit IDs were not released and repositories changed after dataset creation, we use the latest commit as of De-

---

[1]mxbai-embed-large-v1

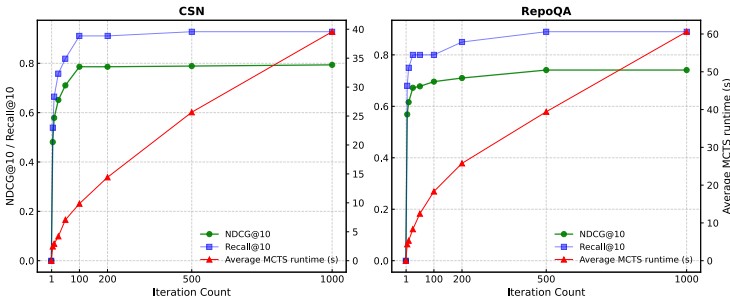

Figure 3: Performance metrics across MCTS iterations for natural language query datasets. Left shows CodeSearchNet NDCG@10, Recall@10, and the average MCTS runtime per iteration across repositories and queries. Right shows RepoQA NDCG@10, Recall@10, and corresponding average runtimes. Both datasets show monotonic improvement followed by convergence, indicating practical iteration ranges for deployment.

cember 31 2023 and re run baselines for consistency. Since all queries here are code entity queries handled directly by Stage 1 we omit text descriptions which are mainly needed for Path 2 MCTS to reduce compute.

### 4.2.2 BASELINES AND RESULTS

Following RepoBench-R setup, the baseline treats import statement snippets as candidate contexts which captures file level linkage. Both RANGER and the baseline use the same rerankers and the same top k protocol to isolate retrieval effects.

Our graph agent improves Accuracy@5, NDCG@5 and MRR@5 across rerankers which shows better localization of fine grained dependencies than file level imports. Pure semantic retrieval performs poorly which supports the need for cross-file graph traversal over linear index search. See Table 2.

Table 2: Performance comparison on the **RepoBench** benchmark for cross-file dependency retrieval.

| Reranker Model | Accuracy@5 | | NDCG@5 | | MRR@5 | |
|---|---|---|---|---|---|---|
| | **RANGER** | **Baseline** | **RANGER** | **Baseline** | **RANGER** | **Baseline** |
| Unixcoder-base (110M) | **0.5446** | 0.4346 | 0.4120 | 0.3075 | 0.3601 | 0.2509 |
| Qwen-3-8B (8B) | **0.5471** | 0.4940 | 0.4120 | 0.3530 | 0.3577 | 0.2919 |

### 4.3 CODE-ENTITY QUERY BASED CODE COMPLETION

#### 4.3.1 DATASET & SETUP

**CrossCodeEval** (Ding et al., 2023) tests cross file code completion across Python, Java, TypeScript and C# using real repositories where the correct continuation depends on cross file context and not just the current file. We use the Python split with 471 repositories, build knowledge graphs from the dataset specified commits, and retrieve cross file context via RANGER. Same as Repobench, for each repository, a code knowledge graph is constructed from the target commit, which is provided in the datasets, without creating text descriptions.

#### 4.3.2 BASELINES AND RESULTS

We compare RANGER against BM25 and several repository level retrievers. **BM25** (Robertson & Zaragoza, 2009) serves as a strong sparse lexical baseline by selecting top-k contexts via term-frequency scoring. **CGM MULTI** (Tao et al., 2025) constructs a one hop ego subgraph around the active file and applies graph aware attention. **RepoFuse** (Cao et al., 2024) fuses analogy contexts with rationale contexts. **RLCoder** (Wang et al., 2024) learns a retrieval policy with perplexity based rewards and a learned stopping rule. **R2C2** (Liu et al., 2024a) assembles repository aware prompts

by selecting candidate snippets with context conditioning. Inspired by RepoFuse, which shows that fusing analogy and rationale contexts improves code generation, we also report **RANGER+BM25** which pairs graph based cross file retrieval with BM25. Since some methods such as RepoFuse and R2C2 use a limit of 4,096 tokens on the retrieved context we also present results with a 4,096 token limit in the Appendix A.3.

Table 3 reports Exact Match and Edit Similarity across DeepSeek Coder 7B and CodeLlama 7B. **RANGER+BM25** achieves the highest Exact Match with DeepSeek Coder 7B and CodeLlama 7B and competitive Exact Match with StarCoder 7B while consistently outperforming BM25. Edit Similarity is mid to strong which reflects the tradeoff between precise dependency localization and lexical similarity. These results underscore the value of explicit graph based retrieval in repository level code completion.

Table 3: Performance comparison of retrieval methods on the **CrossCodeEval** benchmark for Python.

| Retrieval Method | DeepSeek-Coder-7B | | CodeLlama-7B | | StarCoder-7B | |
|---|---|---|---|---|---|---|
| | EM | ES | EM | ES | EM | ES |
| **RANGER + BM25** | **36.27** | 70.77 | **31.68** | 66.91 | 30.80 | 66.03 |
| BM25 | 28.57 | 65.95 | 24.87 | 62.83 | 22.33 | 69.60 |
| CGM-MULTI | 33.88 | 71.19 | 31.03 | **73.90** | **31.00** | 71.66 |
| RepoFuse | 27.92 | 73.09 | 24.80 | 71.05 | 24.20 | 70.82 |
| RLCoder | 30.28 | **74.42** | 26.60 | 72.27 | 25.82 | **72.11** |
| R2C2 | 32.70 | 54.00 | 23.60 | 42.90 | 30.90 | 51.90 |

## 5 CONCLUSION

We introduced RANGER, a repository level agent for graph enhanced code retrieval that handles both *code entity queries* and *natural language queries*. This capability is largely absent from existing code retrieval methods. Our MCTS based graph exploration algorithm, most helpful for natural language queries, uses a bi-encoder for expansion and a cross encoder as the reward. On CodeSearch-Net and RepoQA we surpass strong semantic retrieval systems, including Qwen-3-8B embedding baseline (Wang et al., 2025) ranked number one on MTEB Leaderboard (Muennighoff et al., 2022), while using smaller models for embedding and reranking `mxbai-embed-large-v1` with 335M parameters and `bge-reranker-v2-m3` with 568M parameters. Because cross encoders are more accurate but expensive and often infeasible to apply over the enitre repository, MCTS scores only promising nodes, keeping quality close to exhaustive reranking at lower cost. For repository level completion, where relevant code often lives in other files and is not semantically similar to the query, our graph-guided traversal retrieves the necessary context by following structural relationships rather than embedding proximity alone.

Although RANGER shows strong retrieval performance across multiple benchmarks, several limitations remain. The use of static offline repository graphs limits applicability to dynamic or rapidly evolving codebases where dependencies change frequently. The MCTS stage, while effective for natural language queries, introduces additional inference latency and computational cost that may hinder interactive developer workflows. Node scoring currently depends on cross encoder relevance estimates, which may not be the best reward signal.

Future work will focus on adaptability, efficiency, and evaluation breadth. One direction is incremental graph maintenance that supports live repository updates with minimal recomputation. Another direction is a multi stage retrieval agent in the ReACT style that can combine symbolic Cypher queries with targeted MCTS starting from intermediate graph nodes. This can reduce rollout depth and latency. Learned reward models, including a small language model trained for relevance scoring or reinforcement learning approaches, may offer more robust signals than a fixed cross encoder. At present RANGER supports Python repositories. Since we use the `tree-sitter` library, which is not Python specific and supports many languages, we plan to extend the system to additional languages. Code and resources will be released publicly upon acceptance.

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

# A  APPENDIX

## A.1  PSUEDOCODE OF MCTS ALGORITHM

Below we present the pesudocode for our Monte Carlo Tree Search - based graph traversal algorithm as described in  3.4

---

**Algorithm 1** MCTS-based Graph Traversal Algorithm

---

**Require:**  $q$ (query),   Code graph $\mathcal{G} = (\mathcal{V}, \mathcal{E})$ where each $u \in \mathcal{V}$ has description $D_u$ and embedding $E_u = f_\theta(u)$,   $r \in \mathcal{V}$ (root repository node),   $g_\phi : \mathcal{Q} \times \mathcal{D} \to \mathbb{R}$ (cross-encoder),   $k_{\text{init}}, k_{\text{min}} \in \mathbb{N}$ (initial & min expansion width),   $c > 0$ (UCT exploration),   $\alpha \in [0, 1]$ (score weighting),   $B \in \mathbb{N}$ (retrieval budget),   $T \in \mathbb{N}$ (iterations)

**Ensure:** Ranked node set TopK$(\mathcal{V}_{\text{vis}}, B)$ ordered by retrieval score

**Notation:** For tree node $v$: visits $N_v$, total reward $R_v$, simulation reward $R_v^{(s)}$, simulation visits $N_v^{(s)}$;

$$\text{UCT}(v) = \frac{R_v}{\max(1, N_v)} + c\sqrt{\frac{2 \ln \max(1, N_{\text{parent}(v)})}{\max(1, N_v)}}.$$

Similarity $\text{sim}(E_x, E_y)$ denotes cosine similarity between embeddings.

1: Initialize search tree $\mathcal{T}$ with root $r$; set $N_v \leftarrow 0$, $R_v \leftarrow 0$, $R_v^{(s)} \leftarrow 0$, $N_v^{(s)} \leftarrow 0$ for all $v \in \mathcal{T}$
2: Set expansion width $k \leftarrow k_{\text{init}}$; initialize visited nodes $\mathcal{V}_{\text{tree}} \leftarrow \{r\}$
3: **for** $t = 1$ to $T$ **do**
   **(A) Selection via UCT**
4:    curr $\leftarrow r$
5:    **while** curr has children in $\mathcal{T}$ and not fully expanded **do**
6:       curr $\leftarrow \arg\max_{u \in \text{Children}_\mathcal{T}(\text{curr})} \text{UCT}(u)$
7:    **if** curr is over-visited leaf ($N_{\text{curr}} \geq 2$ and no expandable neighbors) **then**
8:       Traverse up to find expandable ancestor; if none exists, continue to next iteration
   **(B) Expansion**
9:    $\mathcal{C} \leftarrow \text{Neighbors}_\mathcal{G}(\text{curr}) \setminus \mathcal{V}_{\text{tree}}$                          ▷ Non-duplicate children
10:   $\mathcal{S} \leftarrow \{(u, \text{sim}(E_q, E_u)) : u \in \mathcal{C}, E_u \text{ exists}\}$                  ▷ Valid embeddings
11:   **if** $\mathcal{S} = \varnothing$ **then**
12:      Mark curr as fully expanded; **continue**
13:   Sort $\mathcal{S}$ by similarity (descending); $\mathcal{E} \leftarrow \text{TopK}(\mathcal{S}, k)$
14:   Add $\mathcal{E}$ as children of curr in $\mathcal{T}$; $\mathcal{V}_{\text{tree}} \leftarrow \mathcal{V}_{\text{tree}} \cup \mathcal{E}$
15:   Update expansion width: $k \leftarrow \max(k_{\text{min}}, k/2)$                  ▷ Reduce breadth over time
   **(C) Batched Cross-Encoder Simulation**
16:   $\mathcal{P} \leftarrow \{(q, D_u) : u \in \mathcal{E}\}$                          ▷ Query-description pairs
17:   $\mathbf{s} \leftarrow g_\phi(\mathcal{P}) \times 10$                  ▷ Batched cross-encoder inference, scale to [0,10]
18:   rewards $\leftarrow \{u : \text{clamp}(s_u, 0, 10) \text{ for } u \in \mathcal{E}\}$
   **(D) Batched Backpropagation**
19:   **for** each $(u, r_u) \in \{(u, \text{rewards}[u]) : u \in \mathcal{E}\}$ **do**
20:      **for** each $v$ on path from $u$ to $r$ in $\mathcal{T}$ **do**
21:         $N_v \leftarrow N_v + 1$; $R_v \leftarrow R_v + r_u$
22:         If $v = u$: $R_v^{(s)} \leftarrow R_v^{(s)} + r_u$; $N_v^{(s)} \leftarrow N_v^{(s)} + 1$
   **Final Retrieval Score & Ranking**
23: $\mathcal{V}_{\text{vis}} \leftarrow \{v \in \mathcal{T} : N_v > 0\}$
24: For each $v \in \mathcal{V}_{\text{vis}}$, compute retrieval score:

$$s(v) = \alpha \cdot \frac{R_v^{(s)}}{\max(1, N_v^{(s)})} + (1 - \alpha) \cdot \text{sim}(E_q, E_v) \times 10$$

25: **return** TopK$(\mathcal{V}_{\text{vis}}, B)$ sorted by $s(v)$ (descending)

---

## A.2 KNOWLEDGE GRAPH CREATION

Figure 4 illustrates the two-stage repository-level knowledge graph construction process using a simple repository containing `base.py` and `extended.py`.

Stage 1 (File-Level Graph Creation): Individual source files undergo Abstract Syntax Tree (AST) parsing using the tree-sitter library to extract granular code entities. For `base.py`, this creates nodes for the `base` module, `Calculator` class, `add` and `multiply` methods, `format_result` function, and `precision` global variable. Intra-file hierarchical relationships are established through CONTAINS edges (e.g., `base` module contains `Calculator` class and `format_result` function), HAS_METHOD edges (e.g., `Calculator` class contains `add` and `multiply` methods), and HAS_FIELD edges (e.g., `base` module contains `precision` variable).

Similarly, `extended.py` creates nodes for the `extended` module, `Scientific` class, `divide` method, `quick_add` and `demo` functions. Cross-file dependencies that cannot be immediately resolved are represented as temporary placeholder `Import` nodes, shown with dashed lines indicating their eventual resolution targets (e.g., imports of `Calculator`, `precision`, and `format_result` from `base`).

Stage 2 (Repository-Level Graph Consolidation): The system resolves these `Import` nodes through multi-step resolution logic, redirecting all incoming relationships to their actual target entities. For example, the `Scientific` class's inheritance dependency is resolved by establishing a direct INHERITS relationship to the `Calculator` class, and the `quick_add` function's usage of `Calculator` is connected via a USES relationship. After successful edge redirection, the redundant `Import` nodes are deleted, resulting in a unified repository-level knowledge graph that captures both hierarchical structure and cross-file dependencies at the variable level.

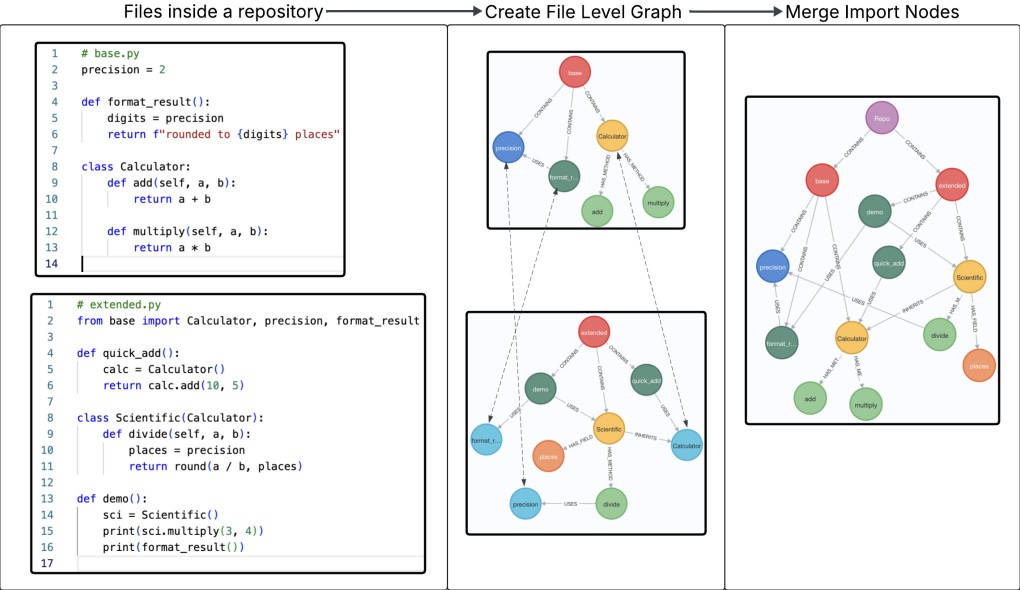

Figure 4: The two-stage graph construction process

## A.3 CROSSCODEEVAL RESULTS WITH RETRIEVED CONTEXT LIMIT

| Retrieval Method | DeepSeek-Coder-7B | | CodeLlama-7B | | StarCoder-7B | |
|---|---|---|---|---|---|---|
| | EM | ES | EM | ES | EM | ES |
| **RANGER + BM25** | **34.03** | 69.48 | 29.89 | 66.32 | 26.94 | 64.01 |
| BM25 | 28.57 | 65.95 | 24.87 | 62.83 | 22.33 | 69.60 |
| CGM-MULTI | 33.88 | 71.19 | **31.03** | **73.90** | **31.00** | 71.66 |
| RepoFuse | 27.92 | 73.09 | 24.80 | 71.05 | 24.20 | 70.82 |
| RLCoder | 30.28 | **74.42** | 26.60 | 72.27 | 25.82 | **72.11** |
| R2C2 | 32.70 | 54.00 | 23.60 | 42.90 | 30.90 | 51.90 |

Table 4: Performance comparison of retrieval methods on the **CrossCodeEval** benchmark for Python with a limit on retrieved context of 4,096 tokens .

## A.4 EXPERIMENTAL PARAMETERS

In this section, we provide the experimental parameters corresponding to the results reported in Section 4.

| Parameter | Specification |
|---|---|
| **RepoBench** | |
| Cypher Generator | hugging-quants/Meta-Llama-3.1-70B-Instruct-AWQ-INT4 |
| **CrossCodeEval** | |
| Cypher Generator | hugging-quants/Meta-Llama-3.1-70B-Instruct-AWQ-INT4 |
| **CodeSearchNet** | |
| Query Embedding (MCTS) | mxbai-embed-large-v1 (335 M Params) |
| Text Description Generation | deepseek-coder-1.3B-instruct |
| MCTS Cross-Encoder | bge-reranker-v2-m3 (568 M Params) |
| MCTS max number of iterations | 200 |
| Total number of 'Module' Nodes in graph | $N$ |
| MCTS $k_{init}$ | $N//2$ |
| MCTS $k_{min}$ | 20 |
| MCTS $c$ (exploration constant) | $\frac{1}{(8\sqrt{\ln(2*N)})}$ |
| MCTS $\alpha$ | 0.5 |
| **RepoQA** | |
| Query Embedding (MCTS) | mxbai-embed-large-v1 (335 M Params) |
| Text Description Generation | deepseek-coder-1.3B-instruct |
| MCTS Cross-Encoder | bge-reranker-v2-m3 (568 M Params) |
| MCTS max number of iterations | 500 |
| Total number of 'Module' Nodes in graph | $N$ |
| MCTS $k_{init}$ | $N//2$ |
| MCTS $k_{min}$ | 20 |
| MCTS $c$ (exploration constant) | $\frac{1}{(\sqrt{\ln(4*N)})}$ |
| MCTS $\alpha$ | 0.9 |

Table 5: Experimental parameters for batch/online processing and evaluation benchmarks.

## A.5 EXAMPLES OF GENERATED CYPHER QUERIES

In this section, we provide examples of Cypher queries generated for different code completion and search tasks across various datasets. Tables 6 and 7 demonstrate the query generation process including entity identification, model reasoning, and the resulting Cypher queries.

Table 6: Natural language query and generated Cypher query example from CodeSearchNet dataset

| **Natural Language Query** | "how to get database table name" |
|---|---|
| **Generated Cypher Query** | |

```
1  MATCH (r:Repo)-[:CONTAINS]->(m:Module {name: 'database'})
2       -[:CONTAINS]->(c:Class)
3  RETURN c.name, c.code
```

Table 7: Entity query, model reasoning, and generated Cypher query for completing the `test_renderables` function in RepoBench dataset

| **Entity Query** | **Input:** `file_name: tests.test_renderables` |
|---|---|
| | Fetch the most important dependencies from the repository to complete the following code: |

```
1   # Copyright (C) 2023  ETH Zurich, Manuel Kaufmann,
2   # Velko Vechev, Dario Mylonopoulos
3   @reference()
4   def test_renderables(viewer: Viewer):
5       grid_xz = np.mgrid[-1.5:1.5:0.3, -1.5:1.5:0.3]
6       n_lines = grid_xz.shape[1] * grid_xz.shape[2]
7       xz_coords = np.reshape(grid_xz, (2, -1)).T
8       line_starts = np.concatenate([
9           xz_coords[:, 0:1],
10          np.zeros((n_lines, 1)),
11          xz_coords[:, 1:2]
12      ], axis=-1)
13      line_ends = line_starts.copy()
14      line_ends[:, 1] = 1.0
15      line_strip = np.zeros((2 * n_lines, 3))
16      line_strip[::2] = line_starts
17      line_strip[1::2] = line_ends
```

| **LLM Thoughts** | Incomplete element identified: function `test_renderables`. |
|---|---|
| **Generated Cypher Query** | |

```
1  MATCH (m:Module {name: 'tests.test_renderables'})
2       -[:CONTAINS]->(f:Function {name: 'test_renderables'})
3  OPTIONAL MATCH (f)-[:USES]->(dep)
4  RETURN DISTINCT dep.name AS name,
5        dep.signature AS signature,
6        dep.code AS code
```

## A.6 SYSTEM PROMPT FOR CROSSCODEEVAL DATASET

The following system prompt is used for generating Cypher queries in the CrossCodeEval evaluation setup. This prompt guides the model to generate precise queries for cross-file dependency analysis while maintaining proper syntax and semantic correctness.

```
1  # Neo4j Cypher Query Expert for Code Dependency Analysis
2
3  You are a Neo4j Cypher query expert. Your task is to generate concise
4  Cypher queries to find ALL cross file dependencies that will help to
5  complete the provided incomplete code based on the provided graph schema.
6
```

```
7   ## Graph Schema:
8   **Nodes**: Repo (name), Module, Class (name, code, signature,
9   module_name), Function (name, code, signature, module_name),
10  Method (name, code, signature, module_name, class)
11  **Edges**: CONTAINS (Repo->Module, Module->Class/Function),
12  HAS_METHOD (Class->Method), INHERITS (Class->Class),
13  USES (All->Dependencies)
14
15  ## Instructions:
16  - **CORRECTNESS**: Use proper Cypher syntax. Ensure each UNION branch
17    in Cypher has a complete MATCH...RETURN with SAME COLUMN NAMES.
18  - **GENERATE MINIMAL QUERIES**: ONLY RETRIEVE THOSE NODES THAT YOU
19    WILL REQUIRE TO COMPLETE THE INCOMPLETE CODE. Use fewest UNION
20    clauses possible.
21  - **MANDATORY**: Return the entire nodes as ***dep*** and their labels
22    as ***label*** in the query. NOTE THE NAMES SHOULD BE 'dep' and
23    'label' ONLY.
24  - **IMPORTANT**: PAY EXTRA ATTENTION TO THE LAST INCOMPLETE LINE, THE
25    FUNCTION/METHOD/CLASS BEING USED IN THE LAST INCOMPLETE LINE, AND
26    TRACE THEM TO WHERE THEY ARE INSTANTIATED/IMPORTED, TO FETCH
27    CORRECT DEPENDENCIES.
28  - **IMPORTANT**: PAY EXTRA ATTENTION TO IMPORT ALIASES, AND ONLY THE
29    GLOBAL VARIABLES BEING USED IN THE LAST INCOMPLETE LINE.
30  - **IMPORTANT**: In the generated cypher query ONLY USE NAMES YOU ARE
31    CONFIDENT ABOUT OR ELSE DON'T USE THEM. For imports, avoid module
32    names as they may differ. It is fine if we get some false positives.
33  - **IMPORTANT**: PAY ATTENTION TO THE PROVIDED GRAPH SCHEMA TO MAKE
34    CORRECT QUERIES.
35
36  ## Input Data Format:
37  Given repo_name: Repository name which can use to identify the Repo
38  Node in the graph.
39  Given file_name: File name which can use to identify the Module Node
40  in the graph.
41  Fetch the most important connected nodes from the graph to predict the
42  next line of the below code:
43  Incomplete code snippet to complete.
44
45  ## Your Task:
46  First provide a brief thought on your decision process, then generate
47  **ONLY THE CYPHER QUERY**.
48
49  **Format:**
50  ```
51  **Thought:** Incomplete element identified: <element_name>
52  (function/method)
53  **Query:**
54  [Cypher query only]
55  ```
56
57  ## Example
58  Given repo_name: /Users/pratik.shah1/work/CrossCodeEval_repos/
59  google_alert-system
60  Given file_name: models.classes
61  Fetch the most important connected nodes from the graph to predict the
62  next line of the below code:
63
64  import numpy as np
65  from poptransformer import ops
66  from poptransformer.layers.layer_norm import BaseLayerNorm
67  from classes import BaseModule as base_module
68
69  class BaseRMSLayerNorm(BaseLayerNorm):
70      def __init__(self, input_size, eps=1e-5, context=''):
71          self.base_object = base_module()
```

```
 72
 73    def collect_bind_layer_weights(self):
 74        weight_key = '.'.join([self.context, 'weight'])
 75        weight_np = self.get_param_from_state_dict(weight_key,
 76                                                   [self.input_size])
 77        self.weight_id = self.add_initialized_input_tensor(weight_np,
 78                                                           weight_key)
 79
 80    def __call__(self, graph, x):
 81        variance_epsilon = ops.constant(graph,
 82                                        np.array(self.eps).astype(np.
                                                float32),
 83                                        'variance_epsilon')
 84        variance = self.base_object.
 85
 86 **Thought:** Incomplete method __call__ in BaseRMSLayerNorm class,
 87 remaining methods are not important. The last incomplete line uses
 88 self.base_object, which calls base_module but that is an ALIAS of the
 89 imported BaseModule class suggesting need for BaseModule. Also need
 90 parent class BaseLayerNorm for inheritance context.
 91
 92 **Query:**
 93 ```cypher
 94 MATCH (r:Repo {name: '/Users/pratik.shah1/work/CrossCodeEval_repos/
 95 google_alert-system'})-[:CONTAINS]->(m:Module)-[:CONTAINS]->
 96 (c:Class {name: 'BaseRMSLayerNorm'})-[:HAS_METHOD]->
 97 (method {name: '__call__'})-[:USES]->(dep)
 98 RETURN DISTINCT dep, labels(dep) as label
 99 UNION
100 MATCH (r:Repo {name: '/Users/pratik.shah1/work/CrossCodeEval_repos/
101 google_alert-system'})-[:CONTAINS]->(m:Module)-[:CONTAINS]->
102 (c:Class {name: 'BaseModule'})
103 RETURN DISTINCT c as dep, labels(c) as label
104 UNION
105 MATCH (r:Repo {name: '/Users/pratik.shah1/work/CrossCodeEval_repos/
106 google_alert-system'})-[:CONTAINS]->(m:Module)-[:CONTAINS]->
107 (c:Class {name: 'BaseLayerNorm'})
108 RETURN DISTINCT c as dep, labels(c) as label
109 ```
```

## A.7 System Prompt for RepoBench Dataset

The RepoBench system prompt is specifically designed for repository-level code completion tasks, with enhanced decision-making logic for identifying incomplete code elements and generating appropriate Cypher queries.

```
 1 # Neo4j Cypher Query Expert for Code Dependency Analysis
 2
 3 You are a Neo4j Cypher query expert. Your task is to generate concise
 4 Cypher queries to find dependencies for code snippets based on the
 5 provided graph schema.
 6
 7 ## Decision Process:
 8 1. **ANALYZE CODE COMPLETENESS**: Check if there's an incomplete element
 9    near the bottom of the code snippet
10 2. **IF COMPLETE**: Use global fallback approach for file-level
11    dependencies
12 3. **TO FIND INCOMPLETE**:
13    - 3a **For collections/lists/dicts**: Missing closing bracket ']',
14        '}', or ')'
15    - 3b **For functions/classes/methods**: Missing body, incomplete
16        signature, or abrupt termination
17 4. **CRITICAL**: Only use visible information. DO NOT GUESS incomplete
18    elements if their definitions aren't clearly shown. **NEVER ASSUME**
```

```
     - if unsure, always use global fallback.
     - **ONLY** identify incomplete elements if you see actual `def`,
       `class`, or variable assignment with collections `[`, `{`
     - Don't identify based on function calls/usage or comments

## Instructions:
- **CRITICAL**: When you find an incomplete function/method/class/
  collection, you MUST identify its name and use the specific template
  for that element - BUT ONLY if the definition is clearly visible
- **NO GUESSING**: If the element definition is not clearly shown, use
  global fallback instead
- **MUST SEE**: Actual `def function():`, `class Name:`, or
  `variable = [` syntax to identify incomplete elements
- **INDENTATION MATTERS**: Pay close attention to indentation to
  distinguish functions (no indent) vs methods (indented) - this is
  crucial for correct queries
- **GENERATE MINIMAL QUERIES**: Use fewest UNION clauses possible
- **MANDATORY**: Return ONLY 'name', 'code', 'signature' attributes
- **IMPORTANT**: Pay attention to complete file path including folder
    names
- **MODULE NODES**: Use 'name' for dotted names, 'local_name' for
  undotted names
- **CORRECTNESS**: Use proper Cypher syntax. Ensure each UNION branch
  in Cypher has a complete MATCH...RETURN with identical column names
  and orders

## Graph Schema:
**Nodes**: Module (name, local_name, code, signature),
Class (name, code, signature, module_name),
Function (name, code, signature, module_name),
Method (name, code, signature, module_name, class),
Field (name, class),
GlobalVariable (name, code, module_name)
**Edges**: CONTAINS (Module->Class/Function/GlobalVariable),
HAS_METHOD (Class->Method), HAS_FIELD (Class->Field),
INHERITS (Class->Class), USES (All->Dependencies)

## Example Queries:

### Example 1 - Incomplete Method
**User Query:**
```
Given file_name: src.alert.interference.reporting.admin.admin
Fetch dependencies for code:
    def get_form_class(self, request, obj=None):
        return ColumnTemplateForm(request)
    def get_client_data(self, request):
```

**Thought:** Incomplete element identified: method `get_client_data`
(based on indentation).

**Query:**
```cypher
MATCH (m:Module {name: 'src.alert.interference.reporting.admin.admin'})
      -[:CONTAINS]->(c:Class {name: 'ColumnTemplateAdmin'})
      -[:HAS_METHOD]->(method {name: 'get_client_data'})
OPTIONAL MATCH (method)-[:USES]->(dep)
RETURN DISTINCT dep.name AS name,
       dep.signature AS signature,
       dep.code AS code
```

## Your Task:
First provide a brief thought on your decision process, then generate
```

```
83   **ONLY THE CYPHER QUERY**.
84
85   **Format:**
86   ```
87   **Thought:** [Incomplete element identified: <element_name> OR
     No incomplete element identified]
89   **Query:**
90   [Cypher query only]
91   ```
92
93   ## User Query:
94   ```
```

## A.8   GRAPH SCHEMA

```
1    # Graph Schema Description
2
3    ## Nodes and Attributes:
4
5    1. **Module**:
6       - **Attributes:**
7          - `name` (String): Dotted module name
8          - `local_name` (String): Local module name (no path)
9          - `embedding` (Vector): Embedding from module description
10         - `description` (String): Summary of the module
11
12   2. **Class**:
13      - **Attributes:**
14         - `name` (String): Class name
15         - `signature` (String): Class signature
16         - `code` (String): Full class code
17         - `module_name` (String): Owning module name
18         - `embedding` (Vector): Embedding from description and
              member_descriptions
19         - `description` (String): High-level summary of the class
20         - `member_descriptions` (String): Descriptions of constituent
              members
21
22   3. **Function**:
23      - **Attributes:**
24         - `name` (String): Function name
25         - `code` (String): Full function code
26         - `signature` (String): Function signature
27         - `module_name` (String): Owning module name
28         - `embedding` (Vector): Embedding from description and
              member_descriptions
29         - `description` (String): High-level summary of the function
30         - `member_descriptions` (String): Descriptions of constituent
              elements
31
32   4. **Field**:
33      - **Attributes:**
34         - `name` (String): Field name
35         - `code` (String): Definition code segment
36         - `class` (String): Owning class name
37         - `description` (String): Summary of the field
38         - `member_descriptions` (String): Details of field usage
39         - `embedding` (Vector): Embedding from description and
              member_descriptions
40
41   5. **Method**:
42      - **Attributes:**
43         - `name` (String): Method name
```

```
44      - `class` (String): Owning class name
45      - `code` (String): Full method code
46      - `signature` (String): Method signature
47      - `module_name` (String): Owning module name
48      - `embedding` (Vector): Embedding from description and
           member_descriptions
49      - `description` (String): High-level summary of the method
50      - `member_descriptions` (String): Descriptions of method members
51
52  6. **GlobalVariable**:
53     - **Attributes:**
54      - `name` (String): Global variable name
55      - `code` (String): Definition code segment
56      - `module_name` (String): Owning module name
57      - `embedding` (Vector): Embedding from description and
           member_descriptions
58      - `description` (String): Summary of the variable
59      - `member_descriptions` (String): Details of variable usage
60
61  7. **Repo**:
62     - **Attributes:**
63      - `name` (String): Repository name
64
65  8. **Import**: (temporary)
66     - **Attributes:**
67      - `name` (String): Imported item name
68      - `module` (String): Source module name
69      - `alias` (String, optional): Alias used in import
70      - `dotted_folder_name` (String, optional): Submodule path
71
72  ## Edges and Relationships:
73
74  1. **CONTAINS**:
75     - **Source:** `Module` or `Repo`
76     - **Target:** `Module`, `Class`, `Function`, or `GlobalVariable`
77
78  2. **HAS_METHOD**:
79     - **Source:** `Class`
80     - **Target:** `Method`
81
82  3. **HAS_FIELD**:
83     - **Source:** `Class`
84     - **Target:** `Field`
85
86  4. **INHERITS**:
87     - **Source:** `Class`
88     - **Target:** `Class` (base class)
89
90  5. **USES**:
91     - **Source:** `Class`, `Function`, `Method`, or `GlobalVariable`
92     - **Target:** `Class`, `Function`, `Method`, or `GlobalVariable`
```

### A.9 PROMPTS FOR GENERATION OF SEMANTIC DESCRIPTION OF ENTITIES

Below are the three prompt templates used to generate high-level and member-specific descriptions for each code entity, as well as the summarization prompt for larger entities (e.g., summarizing the descriptions of all constituent classes, functions, and variables for modules).

### A.9.1 CODE SUMMARIZATION PROMPT

```
1  ### Task: Code Summarization
2
3  Summarize the code at a high level without referencing specific function
```

```
or variable names. Focus on its purpose, how it is implemented, and its
notable features. Use the following format:

**PURPOSE**
Describe what the code is designed to achieve.

**IMPLEMENTATION**
Explain how the code accomplishes its purpose, including general
techniques or components used, without naming exact functions or
    variables.

**KEY FEATURES**
List significant capabilities, design patterns, or behaviors the code
exhibits.

### Programming Language: Python
### Code:
```

### A.9.2 CODE MEMBERS DESCRIPTION PROMPT

```
### Task: Code Members Description

Analyze the Python code and identify important variables (skip temporary
variables and trivial assignments), functions and classes (also function
calls and class instantiations). Use the following format:

name - description

List each important code member with its name followed by a dash and a
*** one-line short description *** of its purpose or functionality.

If no important members are found, respond with: ---None---

***DO NOT REPEAT MEMBERS. YOU CAN CONCLUDE EARLY ONCE ALL MEMBERS ARE
LISTED.***

### Programming Language: Python
### Code:
```

### A.9.3 FILE SUMMARY FROM COMPONENT DESCRIPTIONS PROMPT

```
### Task: File Summary from Component Descriptions

Create a high-level summary of a Python file based on the provided
    component
descriptions. You are not given any code, but only the descriptions of
parts of the code given by various developers. You have to use ALL these
descriptions to summarize the code.

### Guidelines:
1. Do not include any code in your response, or guess the code. Simply
    try and summarize the descriptions provided to you.
2. Focus on the file's overall purpose, architecture, key functionality,
    and key members.
3. If no description is provided simply say 'No description found'.
4. Summarize the purpose of ALL components mentioned in the descriptions.
```

### A.10 LLM USAGE

We used large language models solely for grammar and style polishing. We are fully accountable for all ideas, analyses, and claims, which were authored and verified by us.