# OpenReview forum: "RANGER: Repository-Level Agent for Graph Enhanced Retrieval"
_ICLR.cc/2026/Conference — ICLR 2026 Conference Desk Rejected Submission_

### Official Review · Reviewer_ib2j · 2025-10-26

**Soundness:** 3
**Presentation:** 3
**Contribution:** 2
**Rating:** 4
**Confidence:** 4

**Summary:**

This paper presents RANGER, a repository-level code retrieval agent capable of handling both symbolic (code-entity) and natural language queries within a unified framework.
RANGER first builds a fine-grained repository knowledge graph via AST parsing and cross-file dependency stitching, covering modules, classes, functions, and variables. Each graph node is further enriched with LLM-generated semantic descriptions and embeddings.
At retrieval time, the system employs a dual-stage pipeline:
Code-entity queries are processed via Cypher lookups in a Neo4j-style graph database.
Natural language queries trigger an MCTS-based graph exploration, guided by a bi-encoder for candidate expansion and a cross-encoder for reward-based scoring.
RANGER is evaluated across four diverse benchmarks — CodeSearchNet, RepoQA, RepoBench, and CrossCodeEval — and consistently outperforms strong baselines such as Qwen3-8B, BM25, RepoFuse, and CGM-MULTI in NDCG, Recall, and Exact Match metrics.

**Strengths:**

Novel and unified design combining graph database querying with MCTS-based search.
Strong empirical results across four benchmarks and multiple task types.
Detailed methodology and reproducible description of implementation steps.
Balances accuracy and computational efficiency via selective cross-encoder scoring.
Broad potential impact on retrieval-augmented code reasoning and agentic systems.

**Weaknesses:**

The performance gains, while consistent, are not large enough to be clearly transformative.
The static repository graph limits adaptability to evolving codebases.
MCTS introduces nontrivial latency; no discussion on efficiency trade-offs.
The reward function (cross-encoder scores) lacks learning capability, potentially limiting generalization.
Experiments are restricted to Python; cross-language tests are missing.

**Questions:**

How sensitive is RANGER’s performance to MCTS parameters (c, k, α)?
Could dynamic or incremental graph updates improve scalability?
What is the trade-off between retrieval accuracy and runtime cost?
Have you considered using a learned reward model instead of a fixed cross-encoder?

---

### Official Review · Reviewer_pKpS · 2025-10-28

**Soundness:** 2
**Presentation:** 3
**Contribution:** 1
**Rating:** 2
**Confidence:** 4

**Summary:**

This paper proposes RANGER, a repository-level code retrieval agent that supports both code-entity queries and natural language queries. It first constructs a comprehensive knowledge graph (KG) for the entire repository, enriched with textual descriptions and embeddings to bridge code and natural language. Retrieval with failed Cypher lookups are handled through MCTS-guided graph exploration. RANGER is evaluated across multiple benchmarks, showing strong performance in code search, QA, retrieval, and completion tasks.

**Strengths:**

1. The paper is clearly written and well-structured, making the technical approach accessible and easy to follow.

2. The unified treatment of both code-entity and natural language queries within a single retrieval framework is commendable.

**Weaknesses:**

1. Limited and overstated contributions: (i) The knowledge graph construction closely resembles existing GraphRAG approaches. The use of a graph database is an engineering choice rather than a conceptual contribution. (ii) The only apparent innovation lies in the reward mechanism within the MCTS-guided exploration. (iii) the claimed contribution of a "router" to dispatch query types is misleading: the routing logic is a unified and simple fallback from Cypher lookup failure to MCTS.

2. Missing or unclear implementation details: The KG in Figure 1 includes an edge type labeled "USES", which is not mentioned or defined anywhere in the main body. If this represents variable usage, function calls, or data dependencies, its construction method must be explicitly described.

3. Potentially misleading experimental results: While RANGER uses smaller models and achieves competitive results, the latency of the MCTS process is concerning. Table 1 indicates that MCTS iterations take approximately 15 and 40 seconds, respectively. It is impractical for real-time developer tools and should be explicitly acknowledged and discussed. Additionally, Table 3 omits standalone results for RANGER itself and key baselines such as RepoCoder [1] and DraCo [2]. Moreover, the high Exact Match (EM) but low Edit Similarity (ES) in CrossCodeEval raises concerns: ES is a more user-centric metric, reflecting how much post-completion editing is needed. A high EM with low ES may indicate brittle, exact-string matches that are not practically useful, and this discrepancy warrants deeper analysis.

4. Citation errors: The reference list contains several inaccuracies. For instance, references [2], [3], and [4] have mismatched author lists. While I did not conduct a full audit, this suggests broader issues with citation integrity that could affect the paper’s credibility.

[1] RepoCoder: Repository-Level Code Completion Through Iterative Retrieval and Generation

[2] Dataflow-Guided Retrieval Augmentation for Repository-Level Code Completion

[3] REPOFUSE: Repository-Level Code Completion with Fused Dual Context

[4] GraphCoder: Enhancing Repository-Level Code Completion via Code Context Graph-based Retrieval and Language Model

**Questions:**

See Weaknesses.

---

> ### Author Response · Authors · 2025-11-24
>
> We thank the reviewer for the detailed feedback and address each point in turn.
>
> 1. On “limited and overstated contributions” and the role of the knowledge graph and router
>
>  We agree that graph-based representations for code are not new, and we do not claim that using a graph database alone is a conceptual contribution. Our use of Neo4j is an engineering choice with concrete advantages over in-memory graphs:
>
>  • the repository graph is persistent and can be built once offline and reused without recomputation
>
>  • database-level optimizations (indexes, query planning) enable faster and more scalable retrieval.
>
>
> Our KG construction also differs from prior work (Section 3.2):
>
>  • a two-stage process where file-level graphs are built first and then “stitched” into a repository-level graph
>  • parsing via tree-sitter instead of Python-specific tools such as Jedi/Parso
>
>  • an intermediate JSON representation decoupling language support from the graph backend (new languages need an AST→JSON adapter, new backends a JSON→graph adapter)
>
> We agree that the main conceptual contribution is unified treatment of code-entity and natural-language queries via MCTS-guided exploration over a semantically augmented repository graph, not the mere use of a graph database.
>
> 2. Regarding the “router”
>
> Our routing logic is intentionally simple. We first attempt a Cypher-based retrieval (fast, a few seconds) and fall back to MCTS only if this symbolic path fails or returns no meaningful results. We avoided a separate learned router to remove an extra source of failure and to avoid additional latency. If a router chose the wrong path and that path failed, the system might never try the alternative, whereas in our design MCTS always serves as a fallback when Cypher is unhelpful. Conceptually, the system still routes between symbolic (Cypher) and semantic (MCTS) modes, but with deliberately simple logic.
>
> 3. On the missing definition of the USES edge
>
>  Thank you for pointing this out. The USES edge represents call/usage dependencies between code entities: it connects Class, Function, Method, or GlobalVariable nodes when one entity uses or calls another. This is detailed in Appendix A.8; in the revision, we will also add an explicit definition of USES in Section 3.2 alongside CONTAINS, HAS_METHOD, and INHERITS.
>
>
> 4. On latency, baselines in CrossCodeEval, and EM vs ES interpretation
>
>  We agree that MCTS runtimes on the order of tens of seconds at higher iteration budgets are high for interactive IDE use, and do mention it in the Conclusion. In this paper we focus on retrieval quality in benchmark settings, and the iteration budgets are tuned for accuracy rather than strict real-time constraints.
>
>  Further, in conclusion, we discuss some future work ideas to address this issue:
> A multi-stage ReACT-style retrieval agent that combines fast symbolic Cypher queries with targeted MCTS from intermediate nodes to reduce rollout depth and latency.
>
>
> 4. Regarding missing RepoCoder / DraCo baselines in CrossCodeEval
>
> We followed the Code Graph Model (CGM) paper, which reports CrossCodeEval results for the same subset of methods. To the best of our knowledge at submission time, RepoCoder and DraCo did not publish results on CrossCodeEval in this exact setting, so we compared against methods for which CrossCodeEval numbers were available, using values directly from the original papers.
> For Table 3, we reported RANGER+BM25 because CrossCodeEval is a code completion benchmark where prior work (e.g., RepoFuse) has shown that fusing structured/repository context with BM25 improves EM/ES. Our intention was to be directly comparable to that line of work.
>
> On EM vs ES, we respectfully disagree that “high EM but low ES” necessarily implies brittle or practically useless completions. In our case, as noted in Section 4.3.2, Edit Similarity is mid-to-strong and is close to the best ES across methods. EM is a stricter metric, and high EM indicates that for many instances, the completion exactly matches the ground truth, which is desirable. Because RANGER+BM25 balances cross-file dependency localization (from RANGER) and lexical similarity (from BM25), placing more weight on RANGER tends to improve correctness and EM, sometimes at a cost to ES, while a purely lexical retriever can improve ES but hurt EM if dependencies are not correctly retrieved.
>
> 5. On citation errors
>
>  We appreciate the reviewer catching this issue. There was an ordering/formatting problem in the .bib file that caused some author lists (including [2], [3], and [4]) to be mismatched in the compiled PDF. We will fix the .bib entries for all affected references (including RepoCoder, dataflow-guided retrieval, RepoFuse, and GraphCoder), perform a full pass over the reference list to verify author lists, titles, and venues against the original papers, and recompile to double-check that no inconsistencies remain.

---

> > ### Comment · Reviewer_pKpS · 2025-11-26
> >
> > Thank you for your response. The authors primarily acknowledge but do not address my concerns. I maintain my rating since the limited innovation, efficiency issues, and inadequate baseline still matter.

---

### Official Review · Reviewer_kgfN · 2025-11-03

**Soundness:** 2
**Presentation:** 2
**Contribution:** 2
**Rating:** 2
**Confidence:** 5

**Summary:**

The paper “RANGER: Repository-Level Agent for Graph-Enhanced Retrieval” presents a retrieval agent that unifies symbolic and semantic search for large code repositories. It builds a knowledge graph from Abstract Syntax Trees using the tree-sitter parser, linking modules, classes, and functions through hierarchical and cross-file dependencies. Each node is enriched with LLM-generated descriptions and embeddings, stored in a Neo4j graph database. For queries, RANGER first performs fast Cypher lookups for code-entity queries; if no direct match is found, it switches to a Monte Carlo Tree Search (MCTS) that combines bi-encoder exploration and cross-encoder scoring to locate relevant code. This dual-stage design allows RANGER to handle both structured and natural language queries effectively.

Across benchmarks including CodeSearchNet, RepoQA, RepoBench, and CrossCodeEval, RANGER consistently outperforms strong baselines such as Qwen3-8B embeddings and repository-level retrievers. It achieves higher retrieval precision for natural language queries and better cross-file dependency resolution, while also improving code completion accuracy when combined with BM25. These gains highlight the value of structured graph reasoning over flat vector search.

RANGER’s main strengths are its unified handling of different query types, its efficient MCTS-guided search balancing accuracy and cost, and its persistent, semantically rich graph design. However, it still relies on static, offline graphs and introduces extra computation in MCTS, which may limit use in fast-changing repositories or interactive tools. Despite these trade-offs, RANGER marks a strong step toward scalable, agentic retrieval systems for repository-level code understanding.

**Strengths:**

A key strength of RANGER is that it handles both code-entity and natural language queries within one unified framework. It builds a persistent knowledge graph of the entire repository, capturing module, class, and function relationships while enriching them with LLM-based text descriptions and embeddings. This design allows the system to quickly resolve structured code lookups using Cypher queries while supporting more abstract natural language retrieval through guided graph exploration. Another strength is its Monte Carlo Tree Search (MCTS) mechanism, which balances accuracy and efficiency by using a bi-encoder for exploration and a cross-encoder for fine-grained scoring. This selective scoring method avoids the high cost of full reranking while maintaining near-equivalent quality, helping RANGER find relevant cross-file dependencies that standard embedding searches often miss. Finally, RANGER shows strong empirical results across multiple benchmarks, consistently outperforming established baselines such as Qwen3-8B and RepoFuse on retrieval and completion tasks. Its persistent Neo4j-based graph design also makes it adaptable to new languages and use cases, showing both practicality and consistency across diverse repository-level tasks

**Weaknesses:**

RANGER’s main weaknesses stem from both design choices and scope limitations that echo long-standing issues in graph-based code retrieval research. The most fundamental problem is that it inherits the narrow focus of many prior systems by restricting implementation and evaluation to Python, a language already overrepresented in past graph-based retrieval work. This makes the contribution feel incremental, as numerous earlier methods—such as RepoHyper, RepoGraph, GraphCoder, and DraCo—also relied on Python ASTs and similar dependency modeling. Without demonstrating generalization to statically typed or structurally complex languages like Java, C++, or Go, the system’s real-world relevance remains limited.

Beyond that, RANGER’s static offline graph construction further reduces practicality; it cannot adapt to fast-changing repositories or continuous integration environments where dependencies evolve quickly. The MCTS mechanism, while novel, adds computational overhead and latency that are unsuitable for interactive settings, effectively turning retrieval into a heavy research prototype rather than a production-ready solution. Moreover, the system’s reliance on cross-encoder scoring as the sole reward signal lacks robustness and may overfit to benchmark data instead of scaling to diverse, large repositories.

**Questions:**

1) Since the graph is constructed offline, how does RANGER handle repositories with frequent updates—would incremental graph updates or online synchronization be feasible without full recomputation?
2) The MCTS mechanism adds an additional reasoning layer; have the authors analyzed the latency or cost trade-offs for interactive scenarios like IDE integration, and how might they optimize it for real-time retrieval?
3) The system relies heavily on cross-encoder scoring as the reward function. Would using a learned or adaptive reward model (e.g., reinforcement learning or small LLM fine-tuned for relevance) yield more stable and scalable results?
4) Given that the approach enriches nodes with LLM-generated descriptions, how sensitive is RANGER’s retrieval accuracy to the quality or bias of these generated descriptions?

---

> ### Author Response · Authors · 2025-11-24
>
> We thank the reviewer for the thoughtful analysis and address each point below.
>
> 1. On Python-only implementation and relation to prior graph-based systems
>
>  We agree that the current implementation is limited to Python and that multi-language support is important. We acknowledge this in the conclusion, and since we rely on tree-sitter (which supports many languages), we plan to extend RANGER to additional languages in future work.
>
>
> We also clarify how RANGER differs from prior systems in section 3.2; Many earlier Python-focused approaches (e.g., those based on Jedi, Parso, or other Python-specific analyzers) are tightly coupled to Python tooling. In contrast, RANGER is built on tree-sitter, and the repository-level graph schema (Repo → Module → Class / Function / Field / Variable with CONTAINS, HAS_METHOD, INHERITS, USES, etc.) is deliberately designed to be language-agnostic. While we have only instantiated it for Python so far, the design itself is not tied to that language. We agree that demonstrating RANGER in more languages (e.g., Java, C++, Go) would further strengthen the story and do state this in the Conclusion.
>
>
> 2. Offline graph construction and frequently updated repositories
>
>  We fully agree that static offline graphs limit applicability to rapidly evolving codebases, and hence state it in the conclusion:
>
> “ …the use of static offline repository graphs limits applicability to dynamic or rapidly evolving codebases where dependencies change frequently.”
>
> We also see incremental graph maintenance as a next step and do state “... One direction is incremental graph maintenance that supports live repository updates with minimal recomputation” because:
>
>  • Graph construction is already fast, consisting primarily of AST parsing plus node/edge creation.
>
>  • In a more dynamic setting, one could use git diff (or similar VCS signals) to identify changed files, re-parse only those files, and update the corresponding subgraphs (add/remove nodes and edges, update descriptions/embeddings), avoiding full recomputation.
> We have not yet implemented a full incremental pipeline and do not claim that RANGER already supports this, but it is a future work direction.
>
> 3. MCTS latency and interactive / IDE scenarios
>
>
>  We agree that MCTS adds latency compared to a single dense retrieval pass and do state in the Conclusion. Our focus in this paper we more on retrieval quality in benchmark settings.
>
> “ …The MCTS stage, while effective for natural language queries, introduces additional inference latency and computational cost that may hinder interactive developer workflows.”
>
> Fig. 3 analyzes this trade-off by plotting retrieval metrics and average MCTS runtime as a function of iteration budget, showing that:
>
>  • Increasing the iteration budget improves NDCG/Recall at the cost of additional time.
>
>  • There is an “elbow” region where additional simulations yield diminishing returns, suggesting a practical operating point.
>
> As outlined in the conclusion, we plan to investigate more latency-aware variants, such as a
>
> “... Another direction is a multi-stage retrieval agent in the ReACT style that can combine symbolic Cypher queries with targeted MCTS starting from intermediate graph nodes. This can reduce rollout depth and latency.”
>
> 4. Cross-encoder as reward signal and potential learned reward models
>
>  We agree that relying solely on a fixed cross-encoder as the reward signal may not be optimal, especially for large or diverse repositories, and we note this in the conclusion.
>
> “...Node scoring currently depends on cross-encoder relevance estimates, which may not be the best reward signal.”
>
>  In this initial work we choose a strong off-the-shelf cross-encoder for simplicity and reproducibility, but we view it as a baseline choice rather than a fundamental limitation of the framework.
>
>
> As future work, we highlight:
>
> “... Learned reward models, including a small language model trained for relevance scoring or reinforcement learning approaches, may offer more robust signals than a fixed cross-encoder.”
>
> We will expand this discussion to emphasize that learned/adaptive reward models are an important direction.
>
> 5. Sensitivity to LLM-generated node descriptions
>
>  We agree that sensitivity to LLM-generated descriptions is an important concern. We probe this directly in our natural-language retrieval experiments (Section 4.1) by comparing:
>
>  • Embeddings over raw code chunks.
>
>  • Embeddings over LLM-generated semantic descriptions of the same entities.
>
>
> This setup isolates the effect of descriptive text quality and shows that using LLM-generated semantic descriptions improves retrieval accuracy over using raw code alone, and that RANGER’s MCTS traversal yields further gains on top. At the same time, we acknowledge that if descriptions systematically hallucinate or omit key behaviors, retrieval could degrade, and we will clarify this limitation in the revised version.

---

### Official Review · Reviewer_eG5f · 2025-11-03

**Soundness:** 3
**Presentation:** 2
**Contribution:** 2
**Rating:** 2
**Confidence:** 4

**Summary:**

This paper introduces RANGER, a repository-level retrieval agent for code understanding and retrieval tasks. RANGER builds a comprehensive repository-wide knowledge graph that captures hierarchical and cross-file dependencies down to the variable level and augments each node with natural language descriptions and embeddings. The system integrates a dual-stage retrieval pipeline:
Cypher-based lookups for direct code-entity queries, and MCTS-guided graph exploration for natural language queries with bi-encoder expansion and cross-encoder scoring. Evaluated across CodeSearchNet, RepoQA, RepoBench, and CrossCodeEval, RANGER demonstrates superior retrieval and code completion performance compared to embedding-based and repository-level baselines, including Qwen3-8B, RepoFuse, and R2C2.

**Strengths:**

- address an important problem filling the gap between symbolic code-entity and natural language retrieval at the repository scale.

- novel unified framework integrating graph construction, semantic augmentation with  MCTS-based graph traversal

- performance gains across four benchmarks and multiple metrics (NDCG, Recall, EM, MRR).

- Comprehensive experiments and analyses
- Elaborated related works

**Weaknesses:**

- The main limitation of this work, I think is evaluation. Several key points: RAG level evaluation needs to addressed properly.  1) Repo level retrieval is a key part in SWE tasks where SWE-bench is a very popular codeRAG tasks. 2) Competition level code generation such as on LiveCodeBench is another popular benchmark. I suggest experiments on these two datasets. For SWE any agent AgentLess, Openhands, MiniSWEAgent can be fit with this retriever and other baseline retriever agnet. For LCB, Mapcoder, Self-Reflection, Xolver or any other simple or complex agents can also be used.
- missing baselines: SWE-RANK: https://arxiv.org/abs/2505.07849 could be another baseline as well. When RANGER is combined with BM25 other baselines can also be combined with that.
- Comparison in CodeSearchNet or RepoQA also do not consider strong baselines, rather non retriever language model. Even simple BM25 is not considered.
-MCTS is infact an expensive search method. How do you compare the time complexity? There is no experiments. Fig 3 combining Avg MCTS time with retrieval metics doe not make any sense.
- in Section 4.2 what is the baseleine?
- The finding are not robust, sometimes needs BM25 based hybrid sometimes RANGER only sometime others, no clear conclusion in findings.
- Writing needs improvement and more abalation and analyses.

**Questions:**

See wekanesses

---

> ### Author Response · Authors · 2025-11-24
>
> We thank the reviewer for the summary and for recognizing the importance of repository-level retrieval and our unified graph+MCTS framework. Our work is positioned as a retrieval agent for repository-scale code understanding, and the experiments are designed to isolate the retrieval component rather than evaluate a full end-to-end coding agent.
>
> 1. Evaluation scope, SWE-bench, and LiveCodeBench
>
>
> Our goal is to study repository-level retrieval quality, so we focus on benchmarks where retrieval is the main variable:
>
>  • CodeSearchNet / RepoQA: natural-language–to–code function retrieval.
>
>  • RepoBench-R: cross-file dependency retrieval for code-entity queries.
>
>  • CrossCodeEval: code completion where cross-file context retrieval is the bottleneck.
>
> SWE-bench and LiveCodeBench are less suitable for isolating retrieval. SWE-bench performance depends on planning, tool use, patch generation, and environment interaction. Plugging RANGER into agents such as OpenHands or AgentLess would make the score depend on the rest of the pipeline too. LiveCodeBench is primarily a code generation benchmark, and we do not know of a clean way to turn it into a retrieval benchmark that isolates the retriever from the generator.
>
> We evaluate RANGER on four diverse datasets (CodeSearchNet, RepoQA, RepoBench-R, CrossCodeEval), which we believe gives a robust evaluation, and we agree SWE-bench is a valuable future step.
>
> 2. Missing baselines and SWE-RANK
>
>
> SWE-RANK is designed for SWE-bench-style bug fixing, with an evaluation protocol tightly coupled to that benchmark. Our benchmarks are function-level retrieval and cross-file context retrieval tasks, not bug-fixing or issue-resolution tasks, so directly applying SWE-RANK would require non-trivial re-engineering and re-interpretation.
>
> We agree hybridizing graph-based retrieval with BM25 is a general idea, not unique to our work. In CrossCodeEval we pair RANGER with BM25 because BM25-style lexical retrieval is known to be effective there, and RepoFuse showed that combining repository-level context with BM25 (“analogy + rationale + BM25”) improves generation.
>
> 3. CodeSearchNet / RepoQA baselines and MCTS cost
>
>
> For CodeSearchNet and RepoQA we use a dense retrieval baseline: Qwen-3-8B embeddings are our main semantic baseline and a state-of-the-art embedding model. Existing graph-based systems for code retrieval typically do not support natural-language queries on repository-level graphs, the precise gap RANGER addresses, so for NL queries there is no directly comparable “graph-based NL retriever,” and dense embeddings are the standard choice.
>
> Prior work reports BM25 as generally weaker than dense embedders, which is why we prioritized Qwen-3-8B over BM25 in the main paper. We will add BM25 results for CodeSearchNet and RepoQA and show that RANGER outperforms both BM25 and dense embeddings on these tasks.
>
> We agree that MCTS introduces additional computational cost versus a single-pass dense retriever. In our setting, time complexity depends on graph structure (branching factor and typical path lengths across repositories) and hyperparameters (maximum depth, expansion fan-out, number of simulations/iterations). Rather than give a closed-form bound that hides these dependencies, we report empirical measurements of average MCTS runtime across multiple repositories and will add the complexity equation in the revision.
>
> Fig. 3 shows the trade-off between iteration budget and retrieval quality and highlights an “elbow” region where increasing iterations yields diminishing returns, as both retrieval metrics and runtime increase.
>
> We already mention in the conclusion that MCTS adds computational overhead and will expand the discussion of where higher iteration budgets are acceptable (e.g., offline analysis, batch retrieval) versus where lower budgets are preferable (e.g., interactive IDE settings).
>
> 4. Baseline in Section 4.2
>
>
> For clarity, in Section 4.2.2 we follow the RepoBench-R setup:
>
>  • The baseline treats import statement snippets as candidate contexts capturing file-level linkage.
>
>  • Both RANGER and the baseline use the same rerankers and the same top-k protocol, so differences in Accuracy@5, NDCG@5, and MRR@5 can be attributed to the retrieval mechanism rather than reranking.
>
> 5. Robustness and “no clear conclusion” across benchmarks
>
>
> We appreciate the opportunity to clarify our main story. Across all retrieval-focused benchmarks, RANGER alone (without BM25) is sufficient and performs strongly.
>
>  We use RANGER+BM25 only in CrossCodeEval because it is a code completion benchmark where exact-match-style metrics are sensitive not only to retrieving the correct cross-file context but also to matching repository-specific lexical and stylistic conventions; in that setting, RANGER provides structural and semantic context, while BM25 contributes lexical/style information. For pure retrieval tasks, RANGER alone is sufficient for repository-level retrieval in real code.

---

### Note · Program_Chairs · 2026-01-17
**Submission Desk Rejected by Program Chairs**

The following references in this submission do not refer to real documents and/or have major errors in bibliographic information:

 Yue Zhang, Ge Wang, Shuo Wang, Xiaodong Ke, Shaowei Ma, Hongyu Xu, Ming Zhou, Xu Sun, et al. CodeXGLUE: A machine learning benchmark dataset for code understanding and generation. In NeurIPS Datasets and Benchmarks Track, 2022.
Zexuan Zhong, Xi Victoria Lin, Jiaming Liu, Shuyan Zhou, and Danqi Chen. Retrieval-augmented code generation and understanding. In Advances in Neural Information Processing Systems, 2024.